# Advancing gender equality in global health: What can we learn from successful gender integration across five UN agencies?

Johanna Riha[1]*, T. K. Sundari Ravindran[2], George A. Atiim[3], Renu Khanna[4], Michelle Remme[5]

**1** United Nations University International Institute for Global Health (UNU-IIGH), Kuala Lumpur, Malaysia, **2** Independent Researcher, Thiruvananthapuram, India, **3** Ghana Country Office, World Health Organisation, Accra, Ghana, **4** Centre for Enquiry Into Health and Allied Themes, Mumbai, Maharashtra, India, **5** The Global Fund, Geneva, Switzerland

* johanna.riha@unu.edu

## Abstract

The global nature of on-going crises – climate, economic, social, political – and their impact on health means more than ever a response supported via an effective multi-lateral system is needed. Not only have these crises worsened health inequities but they have also eroded strides advancing gender equality, with detrimental impacts in the health sector and beyond. Despite recent attacks against multilateralism, the United Nations (UN) and its agencies remain strategically well-placed to provide direction and lead the agenda of gender equality in health, drawing on lessons from the past. Through a collaborative practice-based multi-agency study, 14 case studies from across five UN agencies were documented and analysed to identify what has worked institutionally and programmatically to promote gender equality in health over the last 25 years. The outcomes observed reflected the different levels that UN agencies work at and showcased the capabilities and strengths of the UN system in promoting gender equality in health through its operational functions, global agenda-setting work, and institutional processes and structures. In addition, across the case studies five key factors - feminist civil society, robust evidence, leadership support and gender technical expertise, and institutional structures - consistently stood out as necessary elements to leverage opportunities and produce substantial and sustained advances in gender equality in health. These findings offer important lessons of what to foster more of in multilateral and bilateral health organisations as we seek to continue advancing gender equality in health.

## 1. Introduction, rationale and aims

The universal right to "the enjoyment of the highest attainable standard of health is one of the fundamental rights of every human being without distinction of race,

**Data availability statement:** All published UN reports analysed are publicly available. Given some interviews included sensitive information and there is a risk of participant identification, full transcripts cannot deposited in a public repository. However, the analyses and anonymised quotes used can be made publicly available by contacting: iigh-info@unu.edu or the Joint UNU Ethical Review Board at erb@unu.edu.

**Funding:** The author(s) received no specific funding for this work.

**Competing interests:** The authors have declared that no competing interests exist.

religion, political belief, economic or social condition" [1]. Meeting this goal however requires addressing harmful inequalities in power at a social level – be they control over resources, decision-making, access to services, information, and opportunities, all of which impact health and wellbeing. Extensive evidence exists showcasing that one of the primary axes along which harmful power is wielded are socially constructed gendered roles, responsibilities and norms, features that health programmes and institutions often reproduce [2–4]. Far less researched, however, is what works in practice to address these gendered inequalities especially from the perspective of major multilateral actors that work extensively on advancing health equity and gender equality.

The impact of on-going concurrent crises infectious disease outbreaks, alongside economic crises, geopolitical conflicts, climate disasters, and an alarming rise in transnationally coordinated populist movements, is having detrimental effects on advances in health equity and gender equality. The recent intensification of global crises and anti-gender backlash has further raised the urgency to keep gender equality on the global health agenda and for leaders to take corrective action. While there have been some signs of political commitment towards a concerted push for gender equality in health from member states and heads of United Nations (UN) agencies [5,6] recent attacks on multilateralism are endangering these efforts. Now more than ever, it is critical that the UN and its agencies and funds should lead this effort for several compelling reasons.

The UN has played an important agenda-setting role regarding gender equality and women's empowerment since 1975 . A vast resource of experiential knowledge is vested in agencies and actors within the UN system to address gender disparities in health for communities, countries, and regions. Another unique feature is that the UN system is well-positioned to transform evidence and lessons into changes on the ground in the member states through its convening power via the General Assembly (GA), the World Health Assembly (WHA), and other intergovernmental forums. The United Nations Country Team (UNCT), led by the UN Resident Coordinator, ensures coordinated action at the country level. Moreover, the UN system was relatively quick to recognise and respond to the gendered impact of COVID-19 compared to previous health emergencies [7]. This early recognition shows how the UN system can act on lessons learned on how best to advance gender equality in health.

Although a wealth of experiential knowledge exists within UN agencies, evidence of what works has not always been consolidated. Several evaluations have pointed to the lack of knowledge management strategies within many UN Agencies [8]. In addition, formal assessments of gender integration in UN agencies and member states have often been a stock-taking exercise of what has and has not been done, missing opportunities to identify the critical factors necessary for successful gender integration into health programmes and institutional structures [9–12]. In the current climate of major funding cuts to the UN and highly-resources anti-gender movements, it is essential to draw on lessons from the past as we look to strengthen responses for gender equality and health equity in the future.

To fill this major gap at a critical juncture in time, the United Nations University International Institute for Global Health (UNU-IIGH) worked with five UN agencies that operate under a health mandate to document and analyse what has worked institutionally and programmatically to promote gender equality in health over the last 25 years. The study objectives were to:

- Document the types of outcomes UN agencies have been able to achieve in terms of successful programmatic and institutional gender integration in health;

- Identify the determinants that led to the outcomes in successful cases of programmatic and institutional gender integration in health; and

- Distil commonalities and lessons across successful cases to constructively inform future work on gender integration within the UN system and other bilateral and multilateral organisations working in health.

## 2. Methodology and conceptual framework

### 2.1. Ethics statement

Ethical clearance for the study was obtained from the Human Ethics Low Risk Review Committee at Monash University (Project ID: 20317). Written informed consent was obtained from all key informant interviews and workshop participants.

UNU-IIGH collaborated with five UN agencies–the United Nations Joint Programme on HIV/AIDS (UNAIDS) Secretariat, United Nations Population Fund (UNFPA), United Nations Children's Fund (UNICEF), United Nations Development Programme (UNDP), and the World Health Organization (WHO)–working in global health to co-produce practice-based evidence on what works in gender integration in health through an analysis of 14 case studies.

### 2.2. Conceptual framework

Realist synthesis is a theory-driven approach that sets out to understand "what works, where, for whom, why and how" rather than simply evaluate if an intervention works [13]. A realist approach was considered well-suited for this study for several reasons: first, because this study aimed to develop a detailed understanding of the features that led to successful gender integration; second, due to the limited understanding of the pathways leading to successful outcomes in gender integration which is taken into consideration in a realist approach; and third, because of the complexity of social realities that encompass the cases which a realist approach accounts for [13,14]. Underpinning the realist approach is the underlying principle that particular mechanisms, actions, and associated changes lead to desired outcomes within specific contexts [13].

Data collection and analyses were therefore informed by the underlying principle of a critical realist approach, that the relationship between contexts, mechanisms and outcomes needs to be explored to understand how and why an outcome was achieved in some contexts but not others [14]. Consequently, data were gathered not only about successful cases of gender integration but also about the context within which it emerged and factors that facilitated successful implementation and sustenance. The diversity in global health and gender equality mandates across the partner agencies also allowed for exploring how mechanisms operate in different contexts.

### 2.3. Data collection and analysis

Each agency identified a focal contact as the primary link between the research team at UNU-IIGH and the respective agency. Four sequential steps, detailed below, were then used to identify and select case studies and gather and analyse relevant primary and secondary data for the study based on the context, mechanisms and outcomes framework underpinning the realist approach used.

Step one consisted of a detailed review of publicly available and internal agency documents. The purpose was to build an organisational profile of gender integration experiences for each agency and to understand the contextual background necessary to position the case studies of successful gender integration in health. Documents reviewed included agency-specific gender strategies, strategic plans, evaluation reports, annual reports, reports to the executive or governing bodies, knowledge products, tools and guidance related to gender and health, and other web-based resources such as programme descriptions and news stories.

In step two, successful case studies were identified and data were gathered on factors that contributed to successful gender integration. Thirty-nine (39) key-informant interviews were completed across the five agencies between June and September 2020. The purpose of the key informant interviews was to identify successes in gender integration in health, institutionally and programmatically, drawing out explanations on the contextual factors and mechanisms behind sustained successes.

The focal point for the agency identified the sample of relevant key informants from each agency. These consisted of senior management currently working in the headquarters/secretariat and regional and country offices and gender experts with a long track record of gender integration in health currently working in the headquarters/secretariat and/or regional and/or country offices. All key informants barring three fulfilled these criteria. One was a senior manager from UNICEF no longer working with the agency but identified because of their defining role in advancing the gender agenda in the agency; the other was the sole gender expert from the Gender, Equity and Human Rights team of WHO, who had worked with the agency for about two years; and the third was a programme specialist who led some of UNDP's initial gender equality and preparatory work for the 1995 Fourth World Conference on Women in Beijing. Heads of Agencies were not interviewed although they were invited to participate. The sampling method was purposeful, informed and guided by the agency focal point in consultation with the study team, aiming to collect perspectives from a breath of relevant agency stakeholders at headquarter and regional level. The sample size was determined based on reaching saturation of successful cases identified, and the time, resource and travel constraints due to COVID-19.

Two retired UN officials, who had held senior management positions and had extensive expertise in gender equality and women's empowerment, conducted the interviews based on an interview guide co-developed with the study team and the agency focal points. Due to COVID-19-related restrictions on movement, all the interviews were conducted via Zoom with audio or video recording, where additional consent was provided. All interview material, audio, video and transcripts were stored securely, adhering to UNU Policy on Personal Data Protection and United Nations Evaluation Group ethical standards [15,16]. Individual interview transcripts were coded deductively to create a list with brief descriptions of successful cases. Once all the interviews were completed and transcripts coded, a final list of successful cases was compiled for each agency and reviewed by the research team at UNU-IIGH.

Successful cases were defined as follows:

1. Programmatic gender integration – health programmes that resulted in positive gender equality and health outcomes sustained for at least five years. Criteria for defining improvements of health outcomes related to addressing women's, men's and/or gender-diverse people's specific health needs and gender equality outcomes related to altering unequal power relations that are detrimental to health.

2. Institutional gender integration – internal organisational changes that demonstrated improved gender equality, specifically unequal power relations, at the organisational level or significantly enabled programmatic gender integration in health.

The research team at UNU-IIGH then selected a final subset of successful case studies from each agency, covering a range of programmatic and institutional successes. Three case studies were selected for all agencies apart from UNDP where two case studies were selected. This was because although UNDP had several success stories to share from several other sectors, only two suitable ones focused on successes in health. Furthermore, a case study from UNAIDS that

had been sustained for three years was selected as a successful example of programmatic gender integration because it was identified by most of the key informants as an example of success.

For each successful case study, we sought to identify the successful outcome, contextual factors at the global level and within the institution surrounding the emergence of the initiative, and concrete actions (mechanisms) which may have contributed to the successful implementation of the initiative.

To validate our understanding of the circumstances surrounding the success of each case study and fill data gaps, we organised agency-specific virtual workshops. Key informants and other agency staff attended these, including those from regional and country offices. In two instances, ex-staff members were also invited as they could provide additional details related to specific case studies. The proceedings of the workshop were recorded and transcribed.

Step three involved the development of the context-mechanisms-outcome framework and analysis of findings. Once case-study-specific information was validated through agency-specific workshops, analyses were conducted iteratively, evolving and building on new insights and emerging patterns. To do this, we reviewed interview transcripts once again, drew on the background reviews and carried out further web-based research to identify the contextual factors and potential mechanisms for each successful outcome. These sources included internal reports available on agency websites, such as reports to the executive committee, and reports of independent evaluations. This was supplemented by internal documents that the agencies were able to make available. A detailed scrutiny of the information on each successful case study led to the development of a context-mechanisms-outcome framework for analysing the constellation of factors driving successful gender integration initiatives, which was underpinned by the critical realist approach described above. There were three major analytical steps involved:

a) Understanding the specific actors through whom certain actions and associated changes occurred. This representation of relevant actors within the broader conceptual framework provided a more comprehensive picture of who and how specific mechanisms were activated within particular contexts.

b) Determining the triggers that unlocked or catalysed a series of changes that led to the successful outcomes observed. These triggers included a combination of *changes in contextual factors* and *responses* to these changes by specific *actors*.

c) Identifying the *contextual enablers* that created an environment facilitating the triggers to spark change and the *sustaining mechanisms,* such as institutional structures, policies or ways of doing business, helped sustain change over time to lead to successful outcomes.

We then applied this framework to each of the 14 case studies to identify outcomes, actors, triggers, contextual enablers and sustaining mechanisms. Preliminary results based were shared in an internal workshop with a reference group of gender experts. Critical input received from this meeting helped refine and further fine-tune the analysis.

In step four, findings across the case studies were synthesised. Once individual case studies were analysed, case studies were categorised according to type of successful outcome, to understand the types of successful outcomes UN agencies were capable of achieving in programmatic and institutional gender mainstreaming in health. Finally, within each outcome group, patterns across the triggers, contextual enablers and sustaining mechanisms were explored to draw out common factors contributing to successful gender integration.

## 2.4. Positionality of the institute and the research team

UNU-IIGH is an independent global health think tank within the UN system with a mandate to advance the perspectives and needs of states and populations in the Global South to catalyse equitable shifts in power to advance global health. UNU-IIGH works with experts, practitioners, policymakers and academics, including UN agencies. Throughout these interactions, however, it maintains its role as a neutral convener with its staff and associates maintaining their academic

independence. This positionality allows us to critically reflect on the UN system's response to gender integration in health and ensure objectivity and impartial analysis of the evidence collected.

## 3. An analysis of 14 case studies: Lessons from successes in programmatic and institutional gender integration in health

### 3.1. What types of outcomes were achieved?

Details of each of the 14 successful case studies, particularly focusing on what was achieved regarding gender equality, is described in S1 Text. Across the case studies, three overarching types of outcomes were identified, namely outcomes that:

- Empowered women and girls to resist harmful gender norms and practices and advocate for their needs (4 case studies);

- Put gender and health issues on the global agenda (3 case studies);

- Embedded gender equality issues in institutional processes and institutional structures supported gender equality in health programming (7 case studies).

These three types of outcomes reflect the different levels that UN agencies work on and showcase the capabilities and strengths of the UN system. When gender dimensions are successfully integrated into operational functions, agency programmes can empower women, girls and other marginalised groups to resist oppressive gender norms affecting their health. For example, through the work of UNICEF's Menstrual Health and Hygiene (MHH) programme, adolescent girls were empowered to challenge harmful social norms stigmatising menstruation [17–19]. In UNAIDS' work with the Middle East and North Africa's (MENA) Rosa network, women and marginalised groups living with HIV in the MENA region were empowered to resist unequal gender norms and advocate for their needs and rights across a range of national, regional, and global platforms [20–22].

The second outcome category illustrates what is possible when agencies successfully capitalise on their roles in global agenda-setting work, including convening, thought-leadership, evidence generation, advocacy and technical support. The Women's Health Unit, and subsequently the Gender and Women's Health Department within WHO, working in collaboration with external feminist groups and academics for over two decades, managed to get VAWG recognised as a global public health priority through a World Health Assembly resolution passed in 2016. More recently, VAW was included as an outcome indicator in WHO's 13th General Work Plan, with many Member States' health sectors implementing programmes responding to the health consequences of VAW [23,24].

Lastly, outcomes related to successful institutional gender integration showcase the organisational-wide change possible when gender equality is embedded in institutional processes and structures and the positive impact this can have on gender integration in health programmes. For example, PAHO's 2006 *Gender Equality Policy* successfully institutionalised an organisational mandate for gender integration, resulting in gender-mainstreamed health programmes in its Member States [25–27]. UNICEF's first and second Gender Action Plans enabled the agency to systematically operationalise gender integration and include a set of targeted gender priorities in its strategic plan outcomes across its various sectors, including health, since 2014 [28–30].

### 3.2. What were the triggers for the change?

Across the 14 case studies, a specific change in the internal or external context opened up a window of opportunity for gender integration. However, these changes in contextual factors only served as triggers when key actors identified and reacted to these opportunities. Thus, triggers consisted of one or more changes in contextual factors which key actors leveraged to initiate a series of actions related to gender integration. Key actors included senior leadership or in-house

gender experts, with technical expertise and political astuteness, playing a critical role in recognising and leveraging opportunities. Fig 1 illustrates the triggers from each of the case studies.

Examples of successful triggers included:

- In the GBV in humanitarian settings case study, the combination of UNFPA's organisational mandate for humanitarian work through its Strategic Plans and the change in the context when UNFPA became the leader of the GBV Area of Responsibility (AoR) under the Global Protection Cluster of the Inter-Agency Standing Committee, provided the opportunity for the agency to leverage gender expertise of the GBV AoR to mobilise support and buy-in from humanitarian actors across sectors to position GBV as a priority issue [31,32].

- In the UNDP Gender Seal case study, it was UNDP's Strategic Plans (2008–2013) and most recent Gender Equality Strategy (2018–2021) that provided the organisational mandate backing gender equality and women's empowerment in the agency's work. Successive Directors of the Gender and Development team at headquarters, committed to gender equality and women's empowerment, mobilised internal support and funding for the Gender Equality Seal design, pilot and roll out. This included funding for a gender technical expert, who was transferred to the Gender Team at headquarters to lead the development and implementation of the Gender Equality Seal, specifically focusing on country offices [33–37].

- In the case study of the Special Programme for Research and Training in Tropical Diseases (TDR), WHO, a change in the external context, namely significant advances in scholarship vis-à-vis an intersectional approach to gender in public health, motivated staff within TDR to formally incorporate these developments in gender scholarship into their research and training strategies. This evolution in evidence, combined with support from senior leadership, resulted in the formal appointment of an in-house gender expert who positioned the integration of gender in tropical diseases research as an essential part of TDR's research and training mandate, leading to an intersectional gender research strategy [38–41].

In 11 case studies, the changes in contextual factors were favourable—e.g., funding opportunities, the creation of supportive institutional structures, or national government prioritisation of specific health issues. However, in three case studies, the opposite was true. In these instances, changes in contextual factors were associated with unfavourable evaluation reports, which identified significant areas for improvement in gender integration. For example:

- In the UNFPA-UNICEF FGM case study, the change in the context was an evaluation towards the end of the second phase of the programme, which highlighted that gender equality and women's empowerment had not received adequate priority. Despite the programme's acknowledged success in achieving its objectives in the earlier phases—reducing the incidence of FGM—there was a leadership commitment to push forward towards the twin objectives of eliminating FGM and addressing the root causes of harmful practices. As a result, the programme lead at the global level took steps to support changes to the third phase of the programme (2018–2023), which sought to expand the range of interventions aimed at women's and girls' empowerment and to change unequal gender norms [42–44].

### 3.3. What were the 'contextual enablers'?

Although specific triggers precipitated successful outcomes, many contextual 'enablers' had to be in place to activate the triggers and facilitate the successful outcomes observed. These contextual enablers encompass a range of conditions at multiple levels—global, UN system-wide, agency-specific, and national—in which case studies occurred.

For example, positive changes in the context triggered successful gender integration actions within an organisation in 11 out of the 14 case studies. Priority was accorded to gender equality work in organisation-wide strategic plans, resulting in buy-in and commitment from technical departments for gender equality work. Likewise, adverse evaluation outcomes triggered changes in organisational contexts with robust accountability mechanisms through executive-level bodies, which ensured action was taken in response to the evaluations.

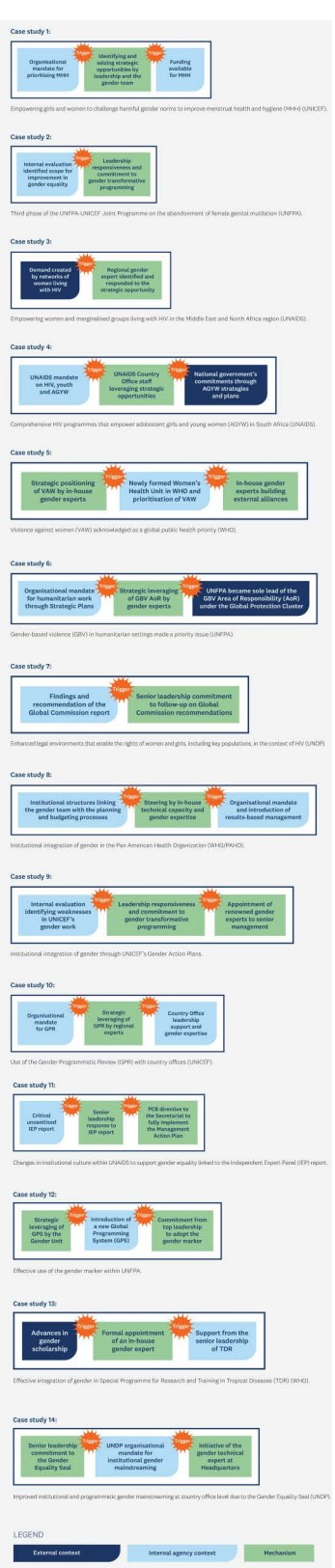

**Fig 1. Triggers precipitating successes in gender integration for each of the 14 case studies.**

Fig 2 summarises the main contextual enablers across the case studies and positions them in relation to triggers and common sustaining mechanisms, all of which collectively result in successful gender integration outcomes. Broadly, contextual enablers can be grouped into external (e.g., global, UN system-wide, and national level circumstances) and internal or agency-specific (e.g., governance structures, processes, and staffing).

External contextual enablers included:

**Feminist civil society movements and campaigns,** which fueled enabling political environments for tackling gender inequalities in health. They contributed significantly to accelerating research and innovation around the differential impact of health programmes on women. For example, the second-wave feminist movements of the 1970s and 80s drew significant attention to VAW as a health and women's rights issue [45]. Global acknowledgement of the harmful effects of FGM and active mobilisation by civil society organisations and the women's and human rights movements at the global level facilitated the UN System's attention and prioritisation of tackling harmful practices affecting women and girls. It led to the launching of the UNFPA-UNICEF Joint Programmeon the Elimination of Female Genital Mutilation [46].

**UN conventions, declarations and resolutions on gender equality**, such as CEDAW in 1979, the 1995 Beijing Conference, the UN Economic and Social Council (ECOSOC) resolution on gender integration of 1997, the 2000 Millennium Development Goals, and most recently, the 2015 Sustainable Development Goals. These secured commitments from national governments to address gender inequalities, including health-related ones, and provided agencies with supportive frameworks to underpin work on gender integration. In some case studies, sector-specific resolutions and political declarations also served as contextual enablers. For example, the UN General Assembly 2016 Political Declaration on Ending AIDS called on governments to reduce new HIV infections among adolescent girls and young women and promote access to tailored, comprehensive HIV prevention services for women and adolescent girls, migrants, and key populations [47].

**Member state support or pressure on particular health issues** created conducive environments within which successful outcomes emerged, particularly in cases of programmatic gender integration. In the FGM case, national government commitments to eliminating FGM, and in some countries, positioning FGM in a larger national agenda of gender equality, sexual and reproductive health and human rights, together with the presence of strong feminist civil society movements [48,49], created enabling environments within which the Joint Programme on the abandonment of FGM could evolve to tackle the unequal power relations, structures and norms that sustain harmful practices. A second example is the priority accorded to gender in PAHO. This was due to the support for gender equality in the region's Member States, which created pressure on the regional office to put gender on the agenda (information from key informant interviews).

**Interagency collaboration leveraging complementary agency strengths** stood out in several cases as a powerful contextual enabler for successful gender integration efforts. One example, is the UNFPA-UNICEF Joint Programme on the Elimination of FGM, which is coordinated and administered by UNFPA but jointly implemented by UNFPA and UNICEF [44]. The programme builds on the respective strengths of the two agencies, particularly UNICEF's extensive operational capacity to work at global, regional, and country levels, as well as a well-resourced Communication for Development Unit within UNICEF and gender expertise across both agencies.

**Donor interests and commitments** for particular issues were critical contextual enablers identified across the case studies. For example, in the UNDP HIV and the Law case study, the Global Fund and PEPFAR emphasised investments in human rights-based programmes for HIV prevention [50]. In addition, there was funding from several other donors (Health Canada, the Norwegian Agency for Development Cooperation, and the Swedish International Development Cooperation Agency) to scale up work on human rights and HIV programmes.

**UN system-wide accountability framework on gender equality and women's empowerment.** The UN SWAP provided a common set of standards that underpinned several successful institutional gender integration efforts, including UNFPA's successful institutionalisation of the gender marker at all levels of the organisation, the institutional integration of gender across UNICEF's work at global, regional and country level, and UNDP's effective use of the Gender Equality Seal for gender integration at country office level [51].

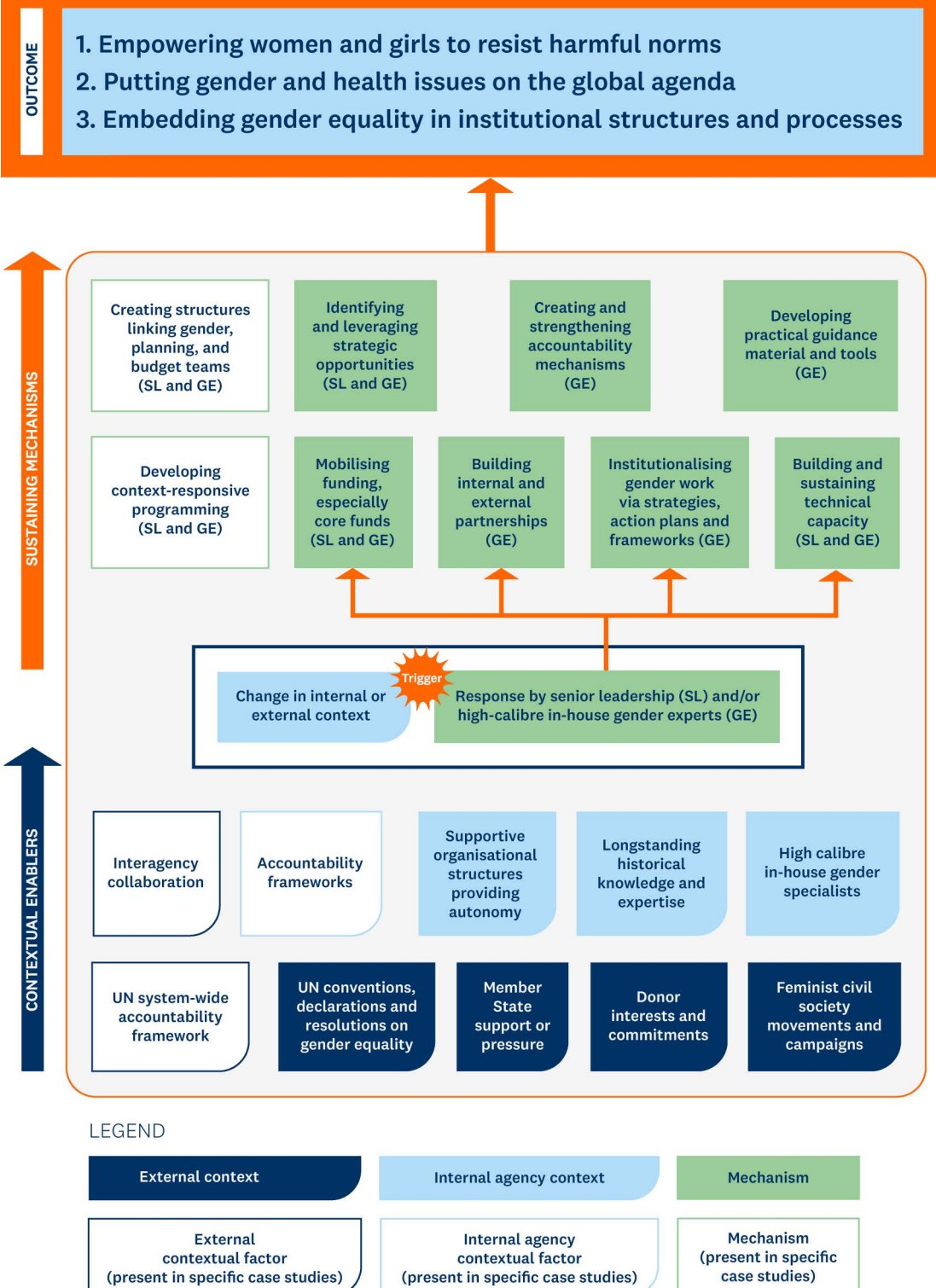

**Fig 2. An illustration of the key triggers, contextual enablers and sustaining mechanisms that led to successes in gender integration across the 14 case studies.**

In addition to the conducive external factors listed above, internal contextual features specific to each agency were critical in creating an enabling environment. These internal contextual features allowed for the activation of triggers and enabled other sustaining mechanisms to occur, ultimately leading to the successful gender integration outcomes observed. Below is a summary of the common internal contextual enablers identified across the case studies:

**Organisational structures that provided autonomy and supportive governance** were crucial in shaping and implementing strategies to strengthen programmatic and institutional gender work. For example, UNICEF' has a streamlined governance structure. The Executive Board is responsible for major decisions, with decision-making authority vested in the Executive Director and Regional and Country Directors. This feature, in combination with the agency's vast network of field offices, has meant that well-designed plans disseminated from headquarters have a high probability of positively influencing the attention paid to gender at all levels of the organization (data from agency-specific workshops).

**High calibre and committed in-house gender specialists**, sufficiently highly positioned at headquarters, regional and country office levels, were critical for success across all cases. A stable and well-staffed gender team was essential to identifying and leveraging strategic opportunities, providing technical support and advice, and developing tools and guidance. For example, in the MHH case study, support from the leadership of the WASH programme and the presence of competent gender expertise in headquarters and regional offices were crucial enablers that strengthened the programme. Gender experts created an evidence base documenting how MHH programmes met several of UNICEF's priorities [52,53]. This strengthened buy-in for the programme across UNICEF.

**Long-standing historical knowledge and expertise** working in specific health areas was a critical contextual enabler across many case studies regarding successful programmatic and institutional gender integration. One example is UNFPA's long track record of promoting health sector responses to GBV in development settings and expertise in gender since 1994. This historical knowledge and expertise enabled UNFPA to advocate for prioritising prevention and services for GBV in emergencies [54].

### 3.4. What were the sustaining mechanisms?

In addition to the triggers and contextual enablers, several crucial actions led by key actors (e.g., senior leadership and in-house gender experts at either headquarters, regional or country level) created and sustained the changes that led to the successful outcomes (see Fig 2). Five main sustaining mechanisms were identified:

**Institutionalising gender equality work.** Across many case studies, developing and implementing institutional frameworks that explicitly prioritised work on gender equality were critical to sustaining gains. These institutional frameworks meant that success was less reliant on individual actors over time, allowing gender integration work to proceed uninterrupted. In the PAHO case, the 2009–2014 and 2015–2019 Plan of Action supported the implementation of the Gender Equality Policy, which had been adopted as a resolution by the PAHO Directing Council in 2005 [55,56]. The Gender Equality Policy and Plan of Action apply to PAHO and Member States and include specific indicators with reporting mechanisms [57]. The introduction of results-based management for the 2014–2019 Strategic Plan provided the gender experts in the Office of Equity, Gender and Cultural Diversity with the opportunity to introduce outcomes and outputs related to gender and other cross-cutting themes into the Programme of Work and Programme Budget [58].

**Creating internal and external accountability mechanisms.** Across the case studies, robust accountability structures were reported as critical mechanisms to accelerate and sustain gender integration efforts linked to the institutionalisation of gender equality work. For example, UNICEF's Gender Action Plan includes Menstrual Health and Hygiene (MHH) indicators, which enforce regular reporting on progress from the MHH programme to the Executive Board and in the GAP annual report [28,59]. In the VAW case study, the Gender, Rights, Advisory Panel of the HRP and SRH Department is an accountability mechanism that monitors and supports the programme [60]. In this case, accountability is also enforced through the 2016 WHA resolution on the Global Plan of Action and by reporting to the WHA against WHO's GPW13 [23,61].

Developing practical guidance material and tools was a fundamental mechanism that sustained success. This included the development of high-quality guidance materials to provide technical support to national stakeholders and to support internal capacity building within agencies. Having competent in-house gender experts who could offer area-specific technical support and guidance was key. In the HIV and the Law case study from UNDP, along with training manuals, capacity-building initiatives focused on human-rights-based approaches to litigation, advocacy on HIV and TB, legal defence, and legal environment assessments. These resources enabled governments and civil society to design tailored country-specific capacity and advocacy activities to demonstrate how legal mechanisms can work for women, girls and key populations in the context of HIV.

Building and sustaining internal and external gender capacity. Linked to providing technical support and guidance is the importance of strengthening internal and external capacity to do gender integration work. For instance, within UNICEF in 2011, core financial resources were committed to creating three senior P5 gender positions at headquarters and seven new P5 senior regional advisors in each of the seven regional offices [62,63]. A new standard issued as guidance to country offices by the Deputy Executive Director recommended country offices with more than $20 million budgets to recruit at least one dedicated Gender Specialist at P3 and P4 levels. For country offices below this budgetary threshold, a gender focal point with at least 20% dedicated time was recommended. This increase in in-house gender architecture contributed to a significant gender integration success within the agency [62].

Mobilising dedicated and sustained funding, especially core funds, was another crucial sustaining mechanism. For instance, in PAHO, while there is no benchmark for budgetary allocation, there is a committed core budget for work on cross-cutting themes, including gender—approximately $12.6 million in 2018–19 and $7 million in 2020–21 [64,65]. Similarly, gender work in TDR received support from core funding to sustain its work (data from agency-specific workshops). Both case studies illustrate how long-term core funding has ensured that work on gender equality is sustained. However, the recent significant reduction in committed core budgets on gender and other cross-cutting themes is a cause for concern.

Building internal and external partnerships were important mechanisms for generating buy-in and support and ensuring the sustainability of gender equality work within health programmes and at an institutional level. These partnerships included other intra-agency technical programmes, national governments, civil society organisations, and affected communities. In the UNICEF Gender Programmatic Review (GPR) case study, the gender advisor and gender specialist looked for opportunities to forge new partnerships through the GPR process. For example, a women's forum in the Kyrgyzstan Parliament and Kosovo's national women's union were involved in the GPR process. Their involvement developed into long-term partnerships, enabling changes on the ground (data from agency-specific workshops).

In the UNFPA GBV in humanitarian settings case study, building sustainability through country and community ownership and partnership with local organisations was another mechanism that facilitated changes on the ground. The GBV programme recognises local organisations' strength in rapid emergency response and knowledge of the contextual specificities. The programme has partnered with government agencies, women's organisations, and women's rights activists to establish a national system for responding to GBV in emergencies (data from key informant interviews).

Although not an explicit mechanism, analyses illustrate the importance of institutional gender integration in achieving successful programmatic integration. For example, successfully integrating gender equality through UNICEF's GAPs provided the institutional mandate for change. It became a trigger for successes initiated through the GPR processes at the country level with support from the regional teams. This successful gender-responsive country programming became an enabling country contextual factor for implementing the MHH programme.

It is important to note that the pathway from a trigger to a successful outcome is not linear, such that once the crucial institutional mechanisms are in place, there is only forward movement on the programmatic front. An organisational mandate, for example, can lapse, strong gender departments can be dissolved, and core budgetary allocations can disappear.

In this regard, accountability mechanisms comprising external actors and alliances with the feminist movement and civil society actors help defend and protect the gains made and resist push-back.

### 3.5. What were the limitations of this study

One of the study's limitations was that it was based on a review of documents, and a limited number of virtual key informant interviews, followed by a virtual analysis workshop. Because of resource, time, and physical constraints due to COVID-19, we could not include gender focal points experts from multiple country offices or any of the partners involved in programme implementation or programme beneficiaries at the country level as key informants. Thus, our information on successes is limited to what was widely perceived to be successes at the headquarters and regional office-level. Also, the scope of the study did not extend to examining the successful outcomes in gender integration in health within the contexts of specific countries and through specific implementing partners. It would be important to replicate this study at each region's regional and country office levels to unpack the necessary ingredients for gender integration into scaled and sustained health programmes that result in positive ground-level changes.The study also does not analyse what did not work and why, which could also serve as a valuable comparison to the successful case studies analysed.

## 4. Discussion and the way forward

This work delineated the key factors necessary to leverage opportunities and create substantial and sustained gains in advancing gender equality in health within the UN system and in other multilateral and bilateral global health organisations. Five key ingredients of success consistently stood out in analyses of the 14 case studies (see Fig 3):

- **The power of leaders and gender experts**

Leadership at the highest levels and gender experts at all levels (headquarters, regional and country level) were crucial for the positive outcomes across all the case studies. In particular, successes were sustained when leadership support was coupled with investment in gender architecture, especially through dedicated core funds. It is important to note that although gender parity in leadership was important, it was insufficient. The skillsets, knowledge, and competence of the individuals appointed were imperative for success.

- **The power of institutional structures**

To translate leadership commitments into concrete action, institutions had to have the organisational infrastructure to take the gender equality agenda forward. Internally, this institutional preparedness involved (1) ensuring direct links between the gender team and the budget/planning teams, bringing the gender integration agenda directly into the decision-making arena; (2) building robust accountability mechanisms at headquarter, regional and country level; (3) making gender equality in health part of the organisation's core business, reflected in all organisational strategy documents as well as programme budgets, with measurable outcome and output indicators. Other institutional features associated with successful examples included decision-making autonomy and adequate financial backing.

- **The power of feminist civil society**

Forming effective partnerships with women's rights organisations was essential for implementing gender-responsive health programmes, ensuring the grounding of programmes in ethical principles and driving local ownership and sustainability. The significant contribution of feminist civil society organisations was particularly notable, where programmes and priorities were jointly defined and shaped through meaningful partnerships between agencies and feminist civil society organisations. However, having representation of civil society alone was not enough. Partnering with *the right* civil society organisations was essential to ensure they provided genuine representation of specific groups and were grounded in feminist ethics.

## The Power of Leaders and Gender Experts

Leaders can catalyse, accelerate and sustain success, by investing in gender architecture across the organisation with dedicated core funds.

- Leverage committed and responsive leadership
- Invest in highly qualified, strategically positioned, gender experts

## The Power of Institutional Structures

Organisational strategies that include gender equality with measurable outcome and output indicators, links between gender teams and budget planning teams, and strong performance and financial accountability mechanisms were gamechangers.

- Create organisational strategies that include gender equality outcome indicators
- Build links between gender teams and budget planning teams
- Strengthen internal performance and financial accountability mechanisms

## The Power of Feminist Civil Society

Feminist civil society expertise and pressure can ensure alignment with local priorities, grounding in ethical frameworks, external accountability and sustainability.

- Build meaningful partnerships in programme design, implementa on, M&E
- Embed feminist civil society groups in institutional governance structures

## The Power of the Collective

Joint interagency collaboration can have real impacts on the ground when comparative advantages of the agencies involved are leveraged.

- Leverage unique normative, operational and coordination roles
- Capitalise on agency-specific agendas and expertise

## The Power of Evidence

Evidence, data and programmatic learning that shows what works (and what the problem is) can drive action and change.

- Build the evidence base on health burden and solutions
- Design programmes to be reflexive and responsive to evidence

**Fig 3. Five ingredients for successful gender integration.**

- **The power of evidence**

Evidence and programmatic learning were central to driving action and change in the case studies. The successful examples illustrated how data and evidence were used to showcase the problem and demonstrate what works. In this regard, evidence-based reflexive learning pushed programme implementers to prioritise approaches that met practical gendered needs and challenged harmful gender norms.

- **The power of the collective**

As several case studies highlight, joint interagency efforts have had real impacts on the ground and offer significant opportunities for advancing gender equality efforts in health. Successful interagency collaboration occurred when the comparative advantages of the agencies involved— their unique agendas, expertise and partnerships with government sectors and different feminist civil society movements— were fully leveraged.

In terms of drawing on published literature to reflect critically on the successes mentioned, to our knowledge, this is the first study of its kind to analyse factors that contributed to successful cases of gender integration in health, programmatically or institutionally, across multiple UN agencies. Where evidence has been published, it is often based on independent evaluations of singular programmes and assessed *whether* a programme was successful rather than *how* or *what enabled* it to be successful. This unique feature of this study therefore sets it apart from the evidence base that exists of formal evaluations of UN programmes focusing on gender integration in health.

### 4.1. Areas for further attention

The process of distilling crucial ingredients for successful programmatic and institutional gender integration brought to the fore several factors within the UN system that appear to work against advancements in gender equality in health. These are listed below as areas for further attention.

**A need to focus on outcomes, not only on processes.** Across the five agencies, there was a widespread focus on processes rather than outcomes in gender integration work. This feature tended to mask the slow progress made in the field and sometimes gave an inflated sense of achievement. For example, where agencies carried out normative and standard-setting work, producing knowledge products and guidance documents was often seen as a success without indicating the uptake, use and impact of these knowledge products on gender equality outcomes in health programmes.

**A need to prioritise a broader range of health and health systems areas.** Across all agencies, programmatic gender integration efforts had a narrow focus on limited health areas and insufficient prioritisation of key health systems areas. Most gender integration efforts centred on GBV and harmful practices, such as FGM, child marriage, and HIV/AIDS. Some agencies have undertaken gender equality work in other health areas like immunisation, WASH, infectious diseases research, and ad hoc work on human resources and universal health coverage. However, these work programmes were few and far between and constituted small fractions of agencies' portfolios. Efforts to integrate gender into major work programmes in areas with large burdens of disease (e.g., non-communicable diseases, emerging infectious diseases) or health system blocks (e.g., service delivery and the health workforce) have proven difficult and unsuccessful to date. A strategic approach prioritising one of these major areas as an entry point for gender integration efforts, backed by adequate financial resources and technical expertise, could have a widespread impact and catalyse a snowball effect for successful programmatic gender integration in other areas.

**A need to build better links between institutional and programmatic gender integration.** This study illustrated the clear links between institutional and programmatic gender integration necessary for successful outcomes. For instance, an organisational mandate, accountability mechanisms, gender architecture and budgetary allocations are essential in enabling programmatic gender integration. However, these findings also highlight the insufficient linkages and learning that occurs the other way around—namely, programmatic gender integration lessons informing institutional efforts. For example, the evidence on what works in transforming gender norms and power at the programmatic/community level is

seldom applied in institutional training on gender equality. This unidirectional view of the relationship between institutional and programmatic gender integration misses ripe opportunities to improve gender equality within organisational structures, processes, and work culture. Ultimately, it negatively impacts the programmatic work as internal organisational mainstreaming is a pre-condition for successful gender integration in operational functions.

### 4.2. Way forward

This particular moment globally is an opportune time to rethink and improve work on gender equality in health, especially in the face of the current health, demographic, financial, social, environmental and political challenges. The global nature of these challenges calls for a response supported through an effective multilateral system, with the UN and its agencies strategically well-placed to lead the agenda of gender equality in healthmaking sure it is not deprioritised during these difficult times.

This study, which has supported agencies to learn from past experiences and build on outcomes where gender has been successfully integrated both institutionally and programmatically into the core business, offers three important recommendations about the way forward:

- **Invest in high-quality, strategically positioned gender experts with decision-making power** at headquarters and regional and country offices. These positions should be core-funded to ensure their sustainability.

- **Combine well-crafted organisational mandates with robust accountability mechanisms** that publicly track and report outcomes, support gender equity goals both institutionally and programmatically, and move funding and spending beyond marker allocations.

- **Identify and seize expected and unexpected changes in contextual factors**, such as exceptionally committed senior leadership, savvy gender experts and leaders, strong donor interest, disruption due to crises, positive shifts in strategic advantage, and organisational restructuring, which present opportunities to create more gender-responsive programmes, put gender and health issues on the global agenda, and strengthen institutional practices that prioritise gender equality in health and other programming.

Now is the time to be bold and meaningfully draw on lessons we can leverage from the past to strengthen advancements for gender equality and health equity in the future.

### Supporting information

**S1 Text.  Summary of the 14 successful case studies.**
(DOCX)

### Acknowledgments

This project would not have been possible without the cooperation of partner agencies. We are indebted to many colleagues at the UNAIDS Secretariat, UNDP, UNFPA, UNICEF, WHO, as well as other key informants, who generously gave their time to share their knowledge, expertise, and decades of experience coordinating gender and health initiatives in the UN system. To maintain confidentiality, we do not list them here by name but wish to acknowledge their substantial contributions in the coproduction of this work. We also thank colleagues from UN Women for their consistent support over the course of this project. We express our deep appreciation to senior consultants, Joanne Sandler, Zineb Touimi-Benjelloun, Aruna Shree Rao, Nandini Oomman and Kathryn Conn, for their significant contributions at various stages of the project. In addition, we thank Avni Amin, Asha George, James Lang, and Franz Wong, for their strategic input and guidance on the study design and/or analyses. We also thank UNU-IIGH colleagues who provided administrative and logistical support for this work: Galila Esam Al-Samawi, Shiau Yun Chong, Amanda Dorsey, Fatima Ghani Gonzalo, Cecilia Hyeinn, Rebecca

Lee, Vidisha Mishra, Aisling Murray, Zaida Orth, Vithiya Sathivelu, and Lavanya Vijayasingham. Finally, we are indebted to Pascale Allotey, for her exemplary leadership over the course of this project and continued commitment to generating policy-relevant evidence that supports UN agencies in their approach to addressing gender inequalities in health.

## Author contributions

**Conceptualization:** Johanna Riha, T. K. Sundari Ravindran, George A. Atiim, Michelle Remme.

**Data curation:** T. K. Sundari Ravindran, George A. Atiim.

**Formal analysis:** Johanna Riha, T. K. Sundari Ravindran, George A. Atiim, Renu Khanna, Michelle Remme.

**Investigation:** Johanna Riha, Michelle Remme.

**Methodology:** T. K. Sundari Ravindran, Michelle Remme.

**Project administration:** Johanna Riha, George A. Atiim, Michelle Remme.

**Supervision:** Johanna Riha, T. K. Sundari Ravindran, Michelle Remme.

**Validation:** T. K. Sundari Ravindran, Renu Khanna.

**Visualization:** Johanna Riha, George A. Atiim.

**Writing – original draft:** Johanna Riha, T. K. Sundari Ravindran, George A. Atiim, Renu Khanna.

**Writing – review & editing:** Johanna Riha, T. K. Sundari Ravindran, George A. Atiim, Renu Khanna, Michelle Remme.

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
