## [Decision Letter · Decision Letter 0]

4 Feb 2025

PGPH-D-24-01960

Advancing Gender Equality in Global Health: What Can We Learn From Successful Gender Integration Across Five UN Agencies?

Dear Dr. Riha,

Thank you for submitting your manuscript to PLOS Global Public Health. After careful consideration, we feel that it has merit but does not fully meet PLOS Global Public Health’s publication criteria as it currently stands. Therefore, we invite you to submit a revised version of the manuscript that addresses the points raised during the review process.

Additional Editor Comments (if provided):

This paper presents an important analysis of the takeaways from successful gender integration across five UN agencies. I read the paper with great interest. In addition, I secured an external review to reconcile conflicting responses from the first two reviewers. Based on both the external review and my assessment, I believe the paper holds merit for publication. However, major revisions are needed to enhance its critical depth and reflexivity.

In addition to the revisions suggested by the reviewers, I have the following concerns:

1. Clarification of Interview Data Usage:

In the methodology section, the authors mention conducting interviews with key informants from several UN agencies. However, there is no clear analysis of these interviews in the findings. If interview data contributed to the results, the authors should specify which sections were informed by the interviews, supported by relevant quotes. If not, references to the interviews should be removed to avoid confusion.

2. Reflexivity and Positionality:

Given that the study was conducted within an institution closely linked to the UN, the authors should include a reflexivity statement. This should clarify how their professional and educational backgrounds, as well as institutional ties with UN agencies, may have influenced the research process, analysis, and interpretation of findings.

3. Enhancing the Discussion Section:

The first part of the discussion largely reiterates findings without deeper analysis. While summarizing key results is useful, this section should critically explore the implications of the findings. For example, while highlighting successes within UN agencies, the authors could contrast these with independent academic evaluations of the real-world impact on women’s lives. Integrating relevant published literature will strengthen this analysis. Additionally, the discussion should explicitly acknowledge the study’s limitations, providing a foundation for future research.

4. Clarification of Ethics Approval:

Although the study is led by the United Nations University International Institute for Global Health, ethical approval was sought from Monash University. This discrepancy requires clarification to ensure transparency in the research process.

5. Declaration of Conflicts of Interest:

The authors should explicitly declare any potential conflicts of interest to maintain the integrity of the publication process.

I hope these suggestions help improve the manuscript’s rigor and critical engagement with the topic.

We look forward to receiving your revised manuscript.

Kind regards,

Naveed Noor, PhD

Academic Editor

Reviewers' comments:

Reviewer's Responses to Questions

**Comments to the Author**

1. Does this manuscript meet PLOS Global Public Health’s publication criteria ? Is the manuscript technically sound, and do the data support the conclusions? The manuscript must describe methodologically and ethically rigorous research with conclusions that are appropriately drawn based on the data presented.

Reviewer #1: Partly

Reviewer #2: No

Reviewer #3: Yes

2. Has the statistical analysis been performed appropriately and rigorously?

Reviewer #1: N/A

Reviewer #2: N/A

Reviewer #3: N/A

3. Have the authors made all data underlying the findings in their manuscript fully available (please refer to the Data Availability Statement at the start of the manuscript PDF file)?

Reviewer #1: Yes

Reviewer #2: No

Reviewer #3: Yes

4. Is the manuscript presented in an intelligible fashion and written in standard English?

Reviewer #1: Yes

Reviewer #2: Yes

Reviewer #3: Yes

5. Review Comments to the Author

Reviewer #1: I am recommending major revisions and have given comments in the reviewer's comments file that I have attached.

The manuscript lacks scientific rigor as the authors have not done adequate literature review. They only have presented UN agencies perspective. The research design and methods that they have use dis not given.

Reviewer #2: Thank you for submitting this paper on the important topic of gender equality. The topic looked promising however there were a few issues that were inconsistent from a research perspective

1. It was not clear how you defined "gender equality" for the study.

2. There was a great deal of high level information and a complex method however there was no tangible connection to on the ground data that would convince a reader that the case studies were successful

3. The post hoc nature of the evaluation and the high level style of presentation makes the work read more like a internal evaluation report than a research report.

Reviewer #3: I commend the authors for their valuable contribution. This study addresses a crucial topic with significant implications for organizations active in the field of global public health, and the authors have presented their findings in a compelling and well-structured manner. The research is timely and contributes valuable insights that can inform both policy and practice in the field.

6. PLOS authors have the option to publish the peer review history of their article (what does this mean? ). If published, this will include your full peer review and any attached files.

**Do you want your identity to be public for this peer review?** For information about this choice, including consent withdrawal, please see our Privacy Policy .

Reviewer #1: No

Reviewer #2: No

Reviewer #3: **Yes: ** Zahra Zeinali

---

## [Decision Letter · Decision Letter 1]

8 May 2025

Advancing Gender Equality in Global Health: What Can We Learn From Successful Gender Integration Across Five UN Agencies?

PGPH-D-24-01960R1

Dear Dr Riha,

We are pleased to inform you that your manuscript 'Advancing Gender Equality in Global Health: What Can We Learn From Successful Gender Integration Across Five UN Agencies?' has been provisionally accepted for publication in PLOS Global Public Health.

Best regards,

Muhammad Naveed Noor, PhD

Academic Editor

Reviewer Comments (if any, and for reference):

Reviewer's Responses to Questions

**Comments to the Author**

1. If the authors have adequately addressed your comments raised in a previous round of review and you feel that this manuscript is now acceptable for publication, you may indicate that here to bypass the “Comments to the Author” section, enter your conflict of interest statement in the “Confidential to Editor” section, and submit your "Accept" recommendation.

Reviewer #1: All comments have been addressed

2. Does this manuscript meet PLOS Global Public Health’s publication criteria ? Is the manuscript technically sound, and do the data support the conclusions? The manuscript must describe methodologically and ethically rigorous research with conclusions that are appropriately drawn based on the data presented.

Reviewer #1: Yes

3. Has the statistical analysis been performed appropriately and rigorously?

Reviewer #1: N/A

4. Have the authors made all data underlying the findings in their manuscript fully available (please refer to the Data Availability Statement at the start of the manuscript PDF file)?

Reviewer #1: Yes

5. Is the manuscript presented in an intelligible fashion and written in standard English?

Reviewer #1: Yes

6. Review Comments to the Author

Reviewer #1: The authors have addressed the comments.

Have added section as per suggestions

7. PLOS authors have the option to publish the peer review history of their article (what does this mean? ). If published, this will include your full peer review and any attached files.

**Do you want your identity to be public for this peer review?** For information about this choice, including consent withdrawal, please see our Privacy Policy .

Reviewer #1: **Yes: ** Narjis Rizvi
